# Isolation of Angola-like Marburg virus from Egyptian rousette bats from West Africa

Brian R. Amman[1], Brian H. Bird[2], Ibrahim A. Bakarr[3], James Bangura[2,4], Amy J. Schuh[1], Jonathan Johnny[3], Tara K. Sealy[1], Immah Conteh[3], Alusine H. Koroma[3], Ibrahim Foday[3], Emmanuel Amara[4], Abdulai A. Bangura[4], Aiah A. Gbakima[5], Alexandre Tremeau-Bravard[3], Manjunatha Belaganahalli[3], Jasjeet Dhanota[2], Andrew Chow[2], Victoria Ontiveros[2], Alexandra Gibson[2], Joseph Turay[4], Ketan Patel[1], James Graziano[1], Camilla Bangura[3], Emmanuel S. Kamanda[3], Augustus Osborne[3], Emmanuel Saidu[3], Jonathan Musa[3], Doris Bangura[3], Samuel Maxwell Tom Williams[3], Richard Wadsworth[3], Mohamed Turay[4], Lavalie Edwin[4], Vanessa Mereweather-Thompson[4], Dickson Kargbo[4], Fatmata V. Bairoh[4], Marilyn Kanu[4], Willie Robert[4], Victor Lungai[4], Raoul Emeric Guetiya Wadoum[4], Moinya Coomber[4], Osman Kanu[4], Amara Jambai[6], Sorie M. Kamara[7], Celine H. Taboy[1], Tushar Singh[8], Jonna A.K. Mazet[2], Stuart T. Nichol[1], Tracey Goldstein ⬤[2]*, Jonathan S. Towner ⬤[1]* & Aiah Lebbie ⬤[3]*

Marburg virus (MARV) causes sporadic outbreaks of severe Marburg virus disease (MVD). Most MVD outbreaks originated in East Africa and field studies in East Africa, South Africa, Zambia, and Gabon identified the Egyptian rousette bat (ERB; *Rousettus aegyptiacus*) as a natural reservoir. However, the largest recorded MVD outbreak with the highest case–fatality ratio happened in 2005 in Angola, where direct spillover from bats was not shown. Here, collaborative studies by the Centers for Disease Control and Prevention, Njala University, University of California, Davis USAID-PREDICT, and the University of Makeni identify MARV circulating in ERBs in Sierra Leone. PCR, antibody and virus isolation data from 1755 bats of 42 species shows active MARV infection in approximately 2.5% of ERBs. Phylogenetic analysis identifies MARVs that are similar to the Angola strain. These results provide evidence of MARV circulation in West Africa and demonstrate the value of pathogen surveillance to identify previously undetected threats.

[1] Viral Special Pathogens Branch, Centers for Disease Control and Prevention, 1600 Clifton Rd. NE, Atlanta, GA 30329, USA. [2] One Health Institute, School of Veterinary Medicine, University of California, 1089 Veterinary Medicine Drive VetMed 3B, Ground Floor West, Davis, CA 95616, USA. [3] Department of Biological Sciences, Njala University, Njala, Sierra Leone. [4] University of Makeni, Makeni, Sierra Leone. [5] Ministry of Technical and Higher Education, New England Ville, Freetown, Sierra Leone. [6] Ministry of Health and Sanitation, Brookfields, Freetown, Sierra Leone. [7] Ministry of Agriculture and Forestry, Brookfields, Freetown, Sierra Leone. [8] Center for Global Health, Centers for Disease Control and Prevention, Freetown, Sierra Leone. *email: tgoldstein@ucdavis.edu; jit8@cdc.gov; alebbie@njala.edu.sl

Marburg virus (MARV), a close relative of the better-known Ebola virus (EBOV), is the founding member of the family *Filoviridae*[1,2] and is known to cause sporadic outbreaks of severe, often fatal disease in humans. There have been 12 known Marburg virus disease (MVD) outbreaks, most recently in 2017 in Uganda[3,4]. The largest MVD outbreak on record occurred in Uige, Angola, in 2005, with 227 deaths out of 252 known cases[5]. This was the highest case-fatality ratio (CFR: 90%) recorded for any large filovirus outbreak, including the 2013–2016 Ebola virus outbreak in West Africa (CFR: 41%,)[6]. A direct link to MARV spillover from bats was not made during the event in Angola. Consistent with the high CFR during the outbreak in Angola, the MARV Angola strain appears to be significantly more virulent than all other MARV strains (Musoke, Ravn, and Ozolin) in experimentally infected non-human primates[7,8]. The Angola outbreak was the only MVD outbreak to originate outside of East Africa; all previous MVD outbreaks occurred in, or originated from, Uganda, Kenya, Democratic Republic of the Congo (DRC), or South Africa ex Zimbabwe[9,10]. Other filoviruses circulating in Africa include the marburgvirus, Ravn virus (RAVV), as well as five ebolaviruses, EBOV, Sudan virus (SUDV), Tai Forest virus (TAFV), Bundibugyo virus (BDBV), and the recently discovered Bombali virus (BOMV)[9,11].

Extensive field studies in Uganda[12–14], DRC[10], Kenya[15], South Africa[16], Gabon[17,18], and Zambia[19,20], as well as experimental infection studies in captive bats in the United States[21,22] and South Africa[23,24], have shown that the cave-dwelling Egyptian rousette bat (ERB; *Rousettus aegyptiacus*) is a primary natural reservoir of MARV. This discovery is consistent with the origins of MVD outbreaks that, when known, have been linked to caves or mines, with MARV most often having spilled over to miners who work underground in known ERB roosting sites, and occasionally to tourists who viewed ERBs too closely[25–28]. Infected ERBs shed MARV in saliva and urine, and the virus can persist for weeks in various tissues, particularly liver, spleen, and lymph nodes[21,22]. Under experimental conditions, MARV can be transmitted directly between ERBs in the absence of arthropod vectors. Furthermore, some infected bats appear to be super-shedders, capable of shedding a disproportionate amount of virus, leading to increased bat-bat transmission in accordance with the Pareto principle[22]. To date, arthropod vectors do not appear to contribute to natural enzootic transmission of MARV among ERBs[10,29–33].

In equatorial Africa, ERBs live in very large, dense colonies sometimes numbering over 100,000 bats[14]. They can breed twice a year, producing thousands of susceptible juvenile bats every six months in a single ERB roost[12,34]. Field studies in Uganda showed that 2–3% of all ERBs are actively infected with marburgviruses (MARV and RAVV) at any one time and that infection levels spike biannually, up to 12% on average, in juvenile bats. Importantly, these seasonal spikes appear to be associated with increased risk of human exposure, as they coincide with >84% of known MARV spillover events to humans[12]. Despite the linkage of ERBs to human MVD outbreaks, attempts to mitigate risk through bat extermination were counterproductive and led to increased levels of active MARV infection in the recolonizing bat population[13].

Since 2007, over 80 distinct MARV genomic sequences and 21 virus isolates have been obtained from tissues of infected wild-caught ERBs, representing every major MARV strain found in MVD outbreaks since 1967, with the exception of the Angola strain[9]. In Gabon, South Africa and Zambia, MARV was detected in ERBs despite no known associated human MVD outbreaks in the country[16–20]. Here, through a combined multi-institutional effort, we report the presence of MARV, including an Angola-like MARV, in ERBs in West Africa.

Importantly, no MVD outbreaks have been reported in Sierra Leone despite the presence of MARV. Our findings highlight the value of engaging with all stakeholders with appropriate messaging that identify and mitigate pathogens of public health concern before recognized spillovers occur. This is in consonant with measures that ensure animal and environmental health. Moreover, it underpins the One Health surveillance approach that recognizes the interconnected relationship between people and other organisms (plants and animals) in a shared environment.

## Results

**Marburg virus detection and isolation.** A total of 1755 bats from 42 species were captured and sampled from 4 districts in Sierra Leone: Moyamba (Kasewe Cave: 8°19′26.80″N, 12°10′36.00″W), Kailahun (Tailu Village: 8°18′27.18″N, 10°30′55.32″W), Koinadugu (Kakoya Cave: 9°41′24.00″N, 11°40′12.00″W), and Kono (Koema Cave: 8°52′12.00″N, 10°48′36.00″W) (Fig. 1). All bat samples were tested for 5 filoviruses (EBOV, TAFV, BDBV, MARV, and RAVV). Of these, 435 bats were identified as ERBs (Kasewe Cave, $n = 186$; Tailu Village, $n = 7$; Kakoya Cave, $n = 131$; Koema Cave, $n = 111$), from which 11 bats (2.5%) tested positive for active MARV infection by virus-specific real-time RT-PCR or consensus RT-PCR. MARV-positive samples included six liver/spleen, five lymph nodes, two oral swabs, one salivary gland, and one whole blood sample (Table 1). MARV isolation was attempted on all PCR positive tissues ($n = 13$) and swabs ($n = 2$), and from those, four virus isolates were obtained from three ERBs caught at Kasewe Cave. Two MARV isolates were obtained from one bat (no. 960), one from the liver/spleen and the other from lymph node, while one isolate was obtained from two other bats (nos. 968 and 1000), each from liver/spleen. Owing to a non-destructive sampling protocol, tissue specimens from ERBs captured at Kakoya and Koema Caves were not available for similar analysis.

**Sequence and phylogenetic analysis.** MARV sequences from small diagnostic NP and VP35 gene fragments were determined from 10 of the 11 PCR-positive bats using an array of sequencing approaches, depending on the institution performing the surveillance and sequence analysis. A synopsis of tissue Ct values, sequences generated, and methodologies used for all qRT-PCR bats is shown in Supplementary Table 1. These MARV sequences were then compared by maximum-likelihood phylogenetic analysis to 128 NP and/or VP35 sequence fragments obtained previously from ERBs or humans in Uganda, DRC, Angola, Gabon, and Kenya. The phylogenetic analysis shows that the Sierra Leone-derived MARV sequences are most closely related to sequences obtained in Gabon and Angola (Fig. 2). In addition, MARV full-length genome sequences were determined by genome walking of MARV RNA extracted from oral swabs and whole blood ($n = 2$), one of which was phylogenetically similar to the Angola-like MARV isolates ($n = 4$) (Fig. 3). Genetic similarities between sequences are shown in Supplementary Tables 2–9. Unexpectedly, MARV isolate sequences from bats nos. 960 ($n = 2$), 968, and 1000 were 100% identical across the full-length virus genome. To rule out cross-contamination during virus isolation, RNA was extracted directly from ERB tissues and approximately 5 kb of MARV RNA was sequenced using an Angola strain-specific tiling and amplification approach. As with the MARV isolate sequences, all tissue-derived MARV sequences from those three bats were 100% identical.

**Marburg virus infection demographics and serology.** Among the 193 ERBs captured at Kasewe Cave and Tailu Village, 140 (72.5%) were juveniles (forearm length <90 mm; Mutere 1968),

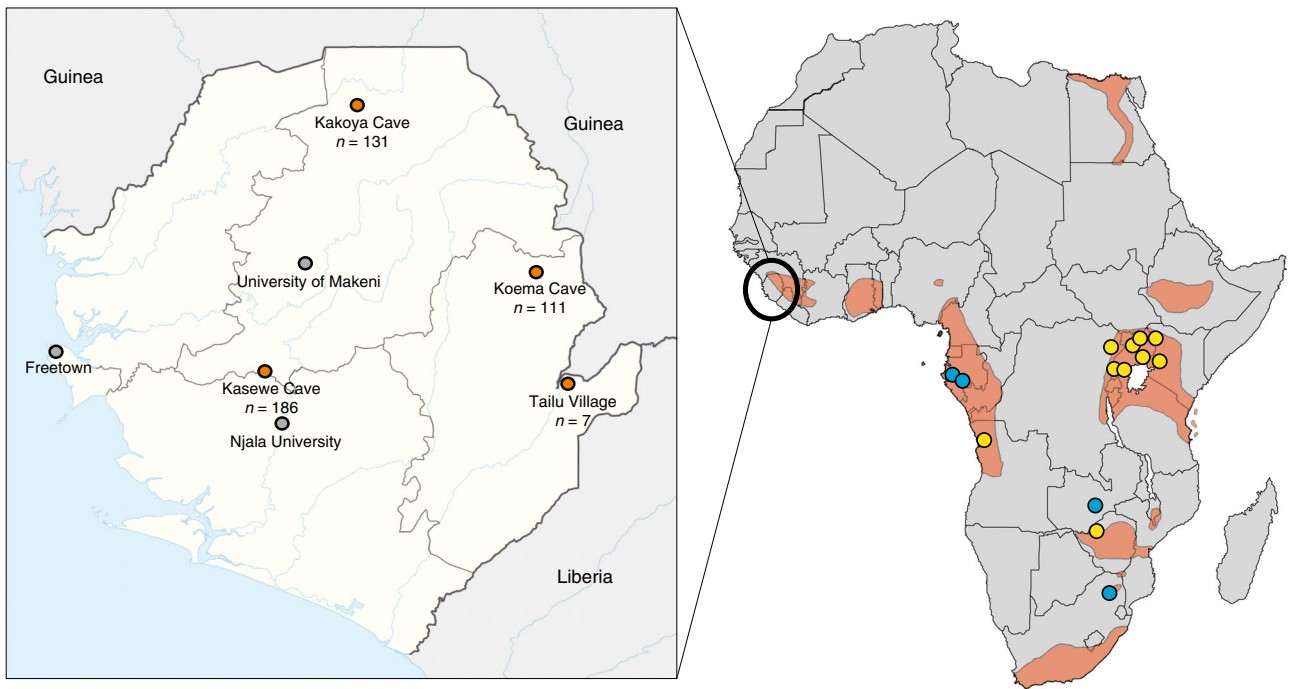

**Fig. 1 Map of Sierra Leone showing bat trapping locations.** Enlarged map shows locations of caves where populations of Marburg virus-(MARV) positive Egyptian rousette bats (ERBs; *Rousettus aegyptiacus*) were discovered (orange circles). The numbers of ERBs captured at each site are shown below the cave name. Shown on the map of Africa are locations of MARV discovery in ERBs without an outbreak (blue circles), known MARV outbreaks (yellow circles), and the fragmented geographic range of the MARV natural reservoir, *R. aegyptiacus* (orange shaded). Image was adapted from base map provided by NordNordWest under Creative Commons Attribution-Share Alike 3.0 Germany license https://creativecommons.org/licenses/by-sa/3.0/de/legalcode.

and 53 (27.5%) were adults. All of the MARV PCR-positive Kasewe Cave ERBs (9/186) were classified as juveniles (4.8%). A total of 242 ERBs were sampled at Kakoya and Koema Caves. Of these, 87 (36%) were juveniles and 155 (64%) were adults. Like the Kasewe Cave and Tailu Village sites, all MARV-PCR positive ERBs (2/242; 0.8%) were juveniles. A significant age bias was detected among MARV-positive bats; all 11 PCR-positive bats were juveniles (Pearson's chi-square; $\chi^2 = 10.341$, $p = 0.001$). No sexual bias with respect to MARV active infection was detected between male ($n = 6$) and female ($n = 5$) PCR-positive bats (Pearson's chi-square; $\chi^2 = 0.003$, $p = 0.954$).

MARV-specific IgG antibody was detected in 24/140 (17.1%) ERBs captured at Kasewe Cave ($n = 136$ serum tested) and Tailu Village ($n = 4$ serum tested). Notably, two of these MARV IgG antibody-positive bats were also positive by qRT-PCR. No sexual bias was observed in MARV-specific IgG antibody-positive ERBs (5/49 [10.2%] female; 19/91 [20.9%] male; Pearson's chi-square; $\chi^2 = 2.555$, $p = 0.110$). Consistent with previous studies of wild-caught ERBs in Uganda[12,14], there was a significant age bias, as 32.4% of adults (12/37) were antibody-reactive to MARV compared to 11.7% of juveniles (12/103; Pearson's chi-square; $\chi^2 = 8.277$, $p = 0.004$). Sera from ERBs captured at Kakoya and Koema Caves were not available for IgG analysis. A summary of ERB qRT-PCR, virus isolation, and ELISA results by sex and age class is shown in Supplementary Table 10.

## Discussion

In this report, we present evidence of active MARV circulation in West African ERBs based on PCR, antibody, and virus isolation data and provide the first report of an Angola-like strain of MARV since it was first detected in humans in 2005. Importantly, this discovery occurred prior to any known MVD outbreak in Sierra Leone and was used to implement evidence-based public

health messaging to at-risk communities about MARV spillover risk. To accomplish this, a comprehensive One Health communications approach leveraging the human, animal, and environmental and emergency health sectors within the Ministries of Health and Sanitation and Agriculture, and Forestry and Food Security along with other international partners was implemented across national, district, and local community levels. Through several engagement meetings with Ministry of Health and Sanitation and with several relevant ministries, departments and agencies, (Ministry of Agriculture Forestry and Food security, Ministry of Local Government, Ministry of Lands, Ministry of Mines and Mineral Resources, Environment and Protection Agency, Office of National Security) over a two-week period, briefing documents including Marburg factsheets, MVD preparedness, detection and response plans were developed and presented at a national conference. This resulted in recommendations for public health outreach, with a team comprised of key stakeholders (government health and agriculture units, universities, development partners and district and local authorities) across the capital city and three of the districts (Moyamba, Koinadugu and Kono). This outreach team conducted initial information sharing events in each community near the ERB colonies followed by regular in-person meetings with traditional community leaders and other local stakeholders to provide key messages related to virus exposure risks and methods to reduce contact with bats. Concerns raised by local communities where bushmeat consumption brings them in contact with bats for livelihood were noted and discussed, and local perceptions about bats were explored in developing options for minimizing exposure risks. As an additional national-level public preparedness measure, MVD has now been included in testing regimens at national laboratories in Sierra Leone.

Marburgviruses have been found in multiple ERB populations across sub-Saharan and South Africa[9]. Though fragmented, the

**Table 1 Summary of MARV infected tissues sampled from *Rousettus aegyptiacus* in Sierra Leone.**

| Collection | Location—district | Bat No. | Species | Sex | Age status | Sample type | MARV sequence | Virus isolated |
|---|---|---|---|---|---|---|---|---|
| Oct. 2017 | Kasewe—Moyamba | 343 | R. aegyptiacus | M | Juv | LN | Yes | No |
| Oct. 2017 | Kasewe—Moyamba | 345 | R. aegyptiacus | M | Juv | Liv/Spl | Yes | No |
| Oct. 2017 | Kasewe—Moyamba | 417 | R. aegyptiacus | F | Juv | LN | Yes | No |
| Sept. 2018 | Kasewe—Moyamba | 940 | R. aegyptiacus | M | Juv | LN | Yes | No |
| Sept. 2018 | Kasewe—Moyamba | 942 | R. aegyptiacus | M | Juv | Liv/SplLN | No | No |
| Sept. 2018 | Kasewe—Moyamba | 960 | R. aegyptiacus | F | Juv | Liv/Spl, LN, SG | Yes | Yes (Liv/Spl, LN) |
| Sept. 2018 | Kasewe—Moyamba | 965 | R. aegyptiacus | F | Juv | Liv/Spl | Yes | No |
| Sept. 2018 | Kasewe—Moyamba | 968 | R. aegyptiacus | F | Juv | Liv/Spl | Yes | Yes |
| Sept. 2018 | Kasewe—Moyamba | 1000 | R. aegyptiacus | M | Juv | Liv/Spl | Yes | Yes |
| Dec. 2017 | Kakoya—Koinadugu | 3960 | R. aegyptiacus | F | Juv | OS, blood | Yes | No |
| Dec. 2017 | Koema—Kono | 4104 | R. aegyptiacus | F | Juv | OS | Yes | No |

*Juv* juvenile, *Liv/Spl* liver/spleen, *LN* axillary lymph node, *SG* salivary gland, *OS* oral swab
Location and characteristics of *Rousettus aegyptiacus* infected with MARV captured in three locations in Sierra Leone with a summary of tissues sampled. Infection status was determined by qRT-PCR and cRT-PCR

geographic range for ERBs extends into West Africa, covering areas of Liberia, Sierra Leone, and Guinea that contain fruiting trees and caves[35]. Therefore, finding 11 MARV positive ERBs from three separate districts (Moyamba, Koinadugu, and Kono) in Sierra Leone is not unexpected, and together with previous field studies[10,12–14,16] supports the evidence that ERBs are the primary MARV natural reservoir. The finding of multiple and diverse MARV genetic lineages simultaneously circulating in geographically distinct locations in Sierra Leone suggests that MARV has been present in West Africa for an extended period of time and is not a recent introduction from other areas of Africa[10,12–14]. Indirect fluorescent antibody data suggested that human MARV infections may have occurred and gone unrecognized in Liberia in the late 1970s, yet due to specificity issues with the indirect immunofluorescence antibody test at that time, the significance of the findings were unclear[36]. Nevertheless, the isolation of genetically identical viruses from 3/9 bats caught at the same cave (Kasewe Cave) was surprising. Of note, all three infected juvenile bats were caught at approximately the same time (within a day of each other). During similar MARV surveillance activities in Uganda from 2007–2012, 21 genetically distinct marburgviruses, including RAVV, were isolated directly from ERBs[12,14], but none were genetically identical to another. In Sierra Leone, we suspect that finding two or more bats simultaneously infected with the same MARV lineage is a consequence of being infected from a single point source, perhaps a supershedder ERB[22] interacting with other bats in a small colony. Moreover, juvenile bats are known to roost together in caves[12], a behavior that may facilitate bat-to-bat transmission from infected to susceptible individuals. In addition, the field teams did not observe evidence of massive ERB colonies at Kasewe, Kakoya, and Koema Caves like those seen in East Africa making multiple infections stemming from one source more likely. The determination that the ERB colonies are comparatively small is based on the lack of widespread fecal deposits on vegetation near the colony entrances, unlike the copious amounts normally seen in East African ERB populations. Future investigations will include

mark-recapture studies to better estimate population sizes at these locations. Overall, the presence of the ERB natural reservoir throughout portions of sub-Saharan Africa implies that marburgviruses could be present in bat populations in many localities with suitable habitat for this species even though no MVD outbreaks have yet been recorded.

The MARV infection data from the four capture sites indicates an age bias towards juvenile ERBs that is consistent with previous studies in Uganda and South Africa[12,16]. Overall, more juveniles were actively infected with MARV, while more adults had antibody reactive to MARV. As with Uganda and South Africa, this is indicative of juveniles having maternal antibody for the first few months after birth, providing protection against MARV infection. That antibody eventually wanes, leaving the older (4-6 months) juvenile cohort susceptible to infection. As the bats get older, the chances of having been infected with MARV increase, leading to the increased prevalence of MARV-specific antibody detected in the adults.

The MARV phylogeny shows that sequences obtained from ERBs in Sierra Leone align most closely with viruses previously found in ERBs in Gabon and DRC from 2006–2009, and in humans in Angola in 2005. The detection of an Angola-like strain is noteworthy because this is the first time it has been identified in ERBs even though all other major marburgvirus lineages, including RAVV, have been detected co-circulating in a single ERB population in Uganda[12–14] or DRC[10]. In that context, the overall genetic diversity detected to date in the West African MARV sequences is comparatively lower and may be a consequence of smaller colony sizes compounded by long-term immunity in previously infected bats. Experimental infection studies of captive ERBs have shown that bats retain immunity to MARV reinfection for up to two years despite diminished antibody levels, suggesting that reinfection is not a major driver of virus persistence in the population[37]. This type of infection dynamic in ERBs would further limit the number of susceptible bat hosts within a colony, thereby potentially limiting the number of virus strains that can co-circulate within an ERB roost. The fact

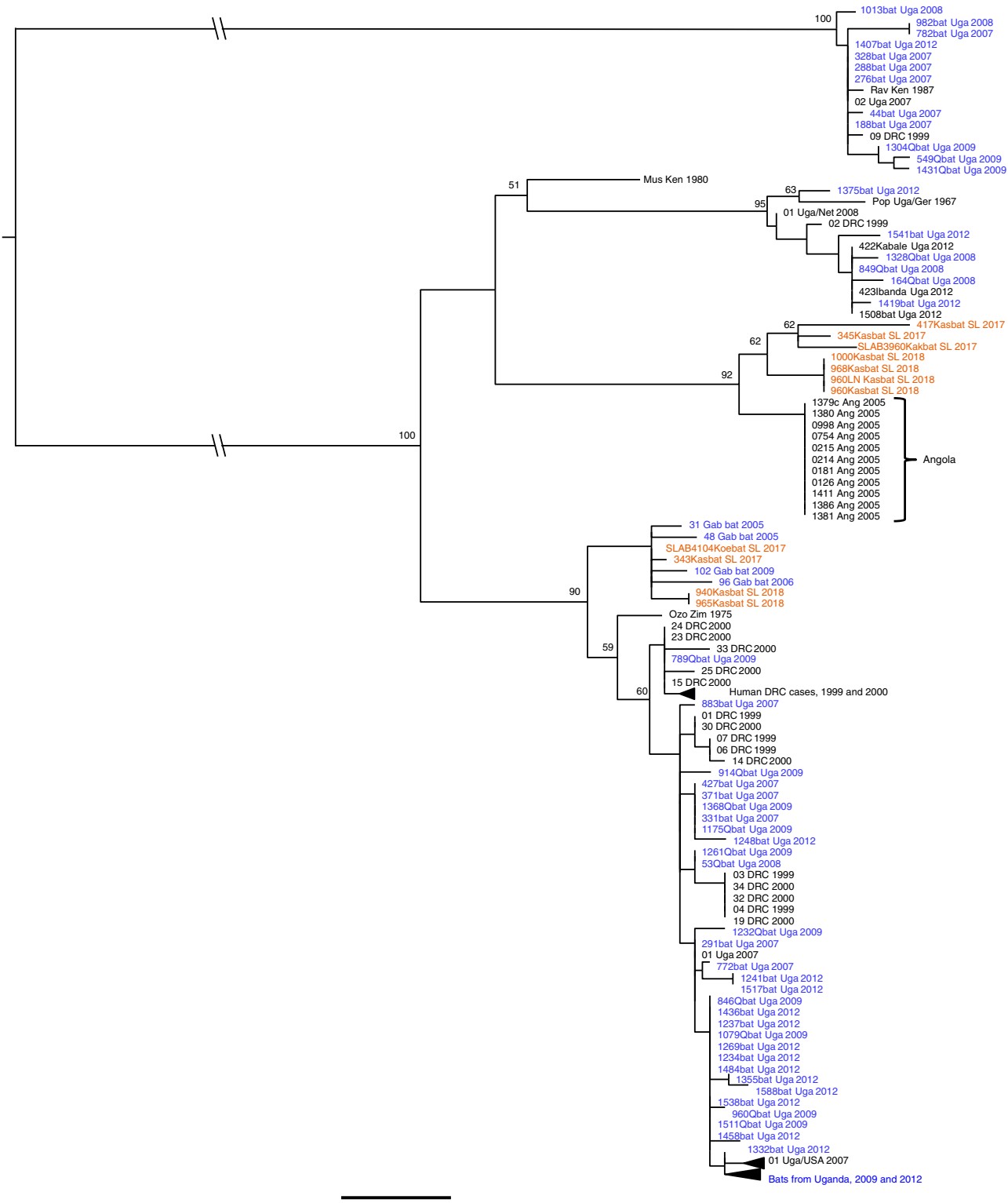

**Fig. 2 Mid-point rooted, maximum-likelihood phylogeny of 128 partial sequence fragments.** Partial and concatenated marburgvirus nucleoprotein (NP) and viral protein 35 (VP35) gene fragments were obtained from *Rousettus aegyptiacus* at three locations in Sierra Leone. Horizontal branch lengths are proportional to the genetic distance between the sequences and the scale at the bottom of the phylogeny indicates the number of nucleotide substitutions per site. Numbers to the left of the nodes represent percent bootstrap values based on 1000 replicates. Only bootstrap values greater than 50% are shown. Sequences in orange represent those generated from the bats in Sierra Leone, sequences in blue represent those generated from bats in Uganda and Gabon and sequences in black represent those generated from human samples. Genbank accession numbers for the Sierra Leone NP and VP35 sequences for all Kasbat SL 2017 and Kasbat SL 2018 sequences are as follows: MN193419—MN193431. The SLAB3960Kakbat SL 2017and SLAB410Koebat SL 2017 NP/ VP35 sequences were pulled from the full-length marburgvirus genome sequences (Genbank accession: MN258361—MN258362).

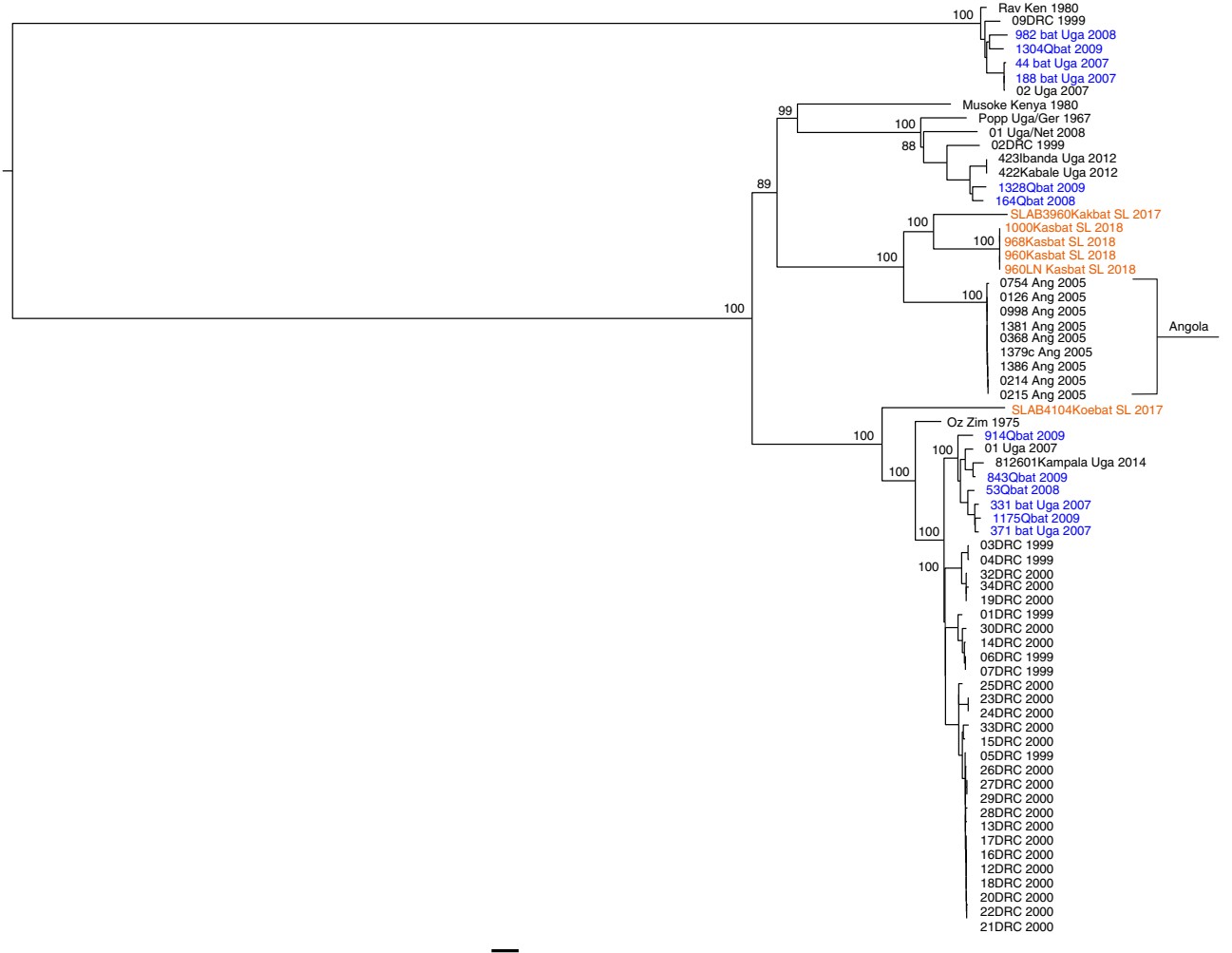

**Fig. 3 Mid-point rooted phylogeny of full-length marburgvirus genomes.** Maximum-likelihood phylogeny of full-length marburgvirus genomes. Horizontal branch lengths are proportional to the genetic distance between the sequences and the scale at the bottom of the phylogeny indicates the number of nucleotide substitutions per site. Numbers to the left of the nodes represent percent bootstrap values based on 1000 replicates. Only bootstrap values greater than 50% are shown. Sequences in orange represent those generated from the bats in Sierra Leone, sequences in blue represent those generated from bats in Uganda and Gabon and sequences in black represent those generated from human samples. Genbank accession numbers for the Sierra Leone full genome sequences for all Kasbat SL 2017 and Kasbat SL 2018 sequences are as follows: MN187403—MN187406. Genbank accession numbers for the SLAB3960Kakbat SL 2017 and SLAB410Koebat SL 2017are MN258361—MN258362.

that the MARV strains detected in Sierra Leonean ERBs are most similar to those seen in other locations on the west coast of Africa (Gabon and Angola) may be reflective of restricted ERB movement and consistent with isolation of ERB populations in Sierra Leone from the larger metapopulation of ERBs across most of central and eastern Africa. One reason for this isolation could be loss of contiguous habitat through degradation of forested lands that bridge the gap between the Congo Basin and West Africa[38].

The clear and unwavering recommendation by the authors of this report is for individuals living and working in close proximity to caves and mines inhabited by ERBs to avoid these bats. Extermination of a reservoir species as a means of zoonotic pathogen control has been shown to be ineffective and can result in higher ratios of active infection[13,39,40]. In one recent example, a Ugandan gold mine was sealed and more than 100,000 ERBs destroyed. Over the course of several years, the bats returned and the prevalence of MARV infection in the bat population more than doubled[13]. This recolonization was soon followed by the largest human MVD outbreak in Ugandan history, centered in a

nearby town[41]. These data show that culling bat populations may lead to increased human health risks and thus should be avoided as a pathogen control measure. Furthermore, as a frugivorous species, ERBs play an extremely important ecological role in forest regeneration by dispersing seeds and facilitating pollination of the fruiting trees they visit on a nightly basis. The ecological benefits of bat activity are critical for the survival of the threatened environment in which they live. Tropical forests in Sierra Leone, Liberia, and Côte d'Ivoire were reported to be most at risk in terms of vulnerability, exposure, and pressure from agricultural expansion[42]. Of those West African countries, Sierra Leone was identified as having the greatest pressure from population and income growth resulting in commodity crop expansion and foreign land investment[42]. With reports of existing vegetative cover in the upper Guinean forests showing losses of nearly 80%[43], ecologically important species like ERBs are crucial to the health and longevity of this fragile ecosystem. Perhaps identifying ERBs as the source of MARV in West Africa can serve as a public deterrent and promote bat avoidance instead of destruction.

## Methods

**Bat capture and processing.** All of the work described herein was performed as a collaborative effort between Njala University, the Centers for Disease Control and Prevention (CDC), the University of Makeni, and the University of California, Davis (UCD) USAID-PREDICT. All animal sampling was performed with permission from the Ministry of Agriculture, Forestry, and Food Security, with approval from both the Institutional Animal Care and Use Committees (IACUC) at the University of California, Davis (protocol number: 16048) and the CDC (protocol number: 2943AMMMULX). The chiropteran taxonomy used in this manuscript follows that of Wilson and Reeder (2005)[44]. Bats were captured at four sites in the Sierra Leone districts of Moyamba (Kasewe Cave), Kailahun (Tailu Village), Koinadugu (Kakoya Cave), and Kono (Koema Cave; Fig. 1). Bats at Kasewe Cave and Tailu Village were captured using mist nets placed at locations around the cave or in a suitable habitat with natural flyways and corridors. Captured bats were placed in breathable cotton bags and transported to a site where they were processed via complete necropsy[45]. Captured bats were humanely euthanized under anesthesia whereupon a cardiac blood sample was obtained. Two oral secretion samples were taken using synthetic poly-tipped swabs (Fisher Scientific, Grand Island, NY, USA or Puritan, Guilford, ME, USA), of which one was placed in virucidal lysis buffer (MagMax; Life Technologies) for PCR analysis and the other in viral transport media for virus isolation. Full necropsies were performed, and visceral tissues (liver, spleen, heart, lung, kidney, salivary gland, axillary lymph node) were collected and either flash frozen in liquid nitrogen for storage or placed in virucidal lysis buffer for inactivation and PCR analysis. When handling bats, all personnel wore appropriate personal protective equipment that included disposable gowns, double gloves including bite gloves (if necessary), face shield and respiratory protection.

Bats at Kakoya and Koema Caves were also captured with mist nets and placed in cotton bags until sampling[11]. Samples were collected in duplicate using non-destructive techniques (venipuncture, oral, and rectal swabs) and placed into viral transport media, frozen at −80 °C or into virucidal lysis buffer (Trizol; Roche Diagnostics) for inactivation and PCR analysis. Bat species field identifications were confirmed using mtDNA barcoding of the cytochrome b and cytochrome oxidase subunit 1 mitochondrial genes[46].

**Statistical analyses.** All statistical analyses were done using SPSS Statistics v25 (IBM Corp, Armonk, NY). Measurements were taken on distinct samples. The ecological data collected at the trapping sites were analyzed for age and sex bias using two-sided Pearson's chi-squared tests.

**Viral detection and sequencing.** Tissue and oral swab samples from bats collected at Kasewe Cave and Tailu Village were analyzed in country at Njala University by quantitative reverse transcriptase PCR (qRT-PCR) targeting marburgvirus protein 40 (VP40—MBGConTaqM F-1: GGA CCA CTG CTG GCC ATA TC, MBGCon TaqM R-3Rev-1: GAG AAC ATl TCG GCA GGA AG; Probes: MBG5313-Prb: 56-FAM-ATC CTA AAC-ZEN-AGG CTT GTC TTC TCT GGG ACT T-3IABkFQ, MBG5313-PrbRav: 56-FAM-ATC CTG AAT-ZEN-AAG CTC GTC TTC TCT GGG ACT T-3IABkFQ). Total RNA was extracted from tissue and oral swabs. RNA from tissues, approximately 100 mg of tissue was placed in 2 mL grinding vials (OPS Diagnostics, Lebanon, NJ, USA) with 1 mL of MagMax lysis buffer concentrate (Ambion MagMAX, Applied Biosystems, Foster City, CA, USA) and homogenized. 125 µL of tissue lysate was then mixed with 75 µL of 100% isopropanol; for oral swab RNA, 500 µL of swab eluate in lysis buffer was extracted on the MagMax Express-96 deep-well magnetic particle processor using pre-loaded protocols. A commercially available eukaryotic 18S rRNA primer/probe assay (Applied Biosystems, Grand Island, NY, USA) was used according to manufacturer's instructions for an extraction control.

Sequences were obtained using products generated from RT-PCR for MARV VP35 (Round 1: MBGVP35-F1: GCTTACTTAAATGAGCATGG, MBGVP35-R1: AGIGCCCGIGTTTCACC; Round 2: MBGVP35-F3: CAAATCTTTCAGCTAA GG, MBGVP35-R2: TCAGATGAATAIACACAIACCCA) and NP amplicons (Round 1: MBG704F1+: GTAAAYTTGGTGACAGGTCATG, MBG1248R1-: TCT CGTTTCTGGCTGAGG; Round 2: MBG719F2+: GGTCATGATGCCTATGACA GTATCAT; MBG1230R2-: ACGGCIAGTGTCTGACTGTGTG)[5,18]. For the first round of PCR, 10 µL of extracted RNA was used. The second round of PCR used 7 µL of the first round PCR reaction. PCR amplicons were analyzed on 2% E-gel. PCR reactions were purified using Ampure beads and quantitated using Qbit dsDNA HS Assay Kit on a Qubit 3.0 fluorimeter. The MinION platform was used with R9 flow cells (FLO-MIN107) and the 1D Barcoded ligation sequencing kit (1D SQK-LSK108 with the EXP-NBD103 Native Barcoding kit 1D) for sequencing of amplicons. Sequencing libraries were constructed and performed according to manufacturer's instructions. Base calling was performed using ONT Albacore sequencing pipeline software (version 2.1.10). Reads were mapped to the reference genome by means of bwa mem using the -x ont2d flag. Reads were then converted to BAM format using SAMtools view. SAMtools was then used to sort the aligned BAM files. Consensus sequences were obtained using alignment-base consensus generation in Geneious (version 11.1.4). Each consensus sequence was manually reviewed and edited for base calling errors (specifically in homopolymer regions). Primer sequences used to generate the amplicon were manually trimmed.

Bats captured at Kakoya and Koema Caves were processed at the University of Makeni. Samples were then shipped to UC Davis where total RNA was extracted using Direct-Zol RNA columns (Zymo Research Corp), and cDNA prepared using Superscript III (Invitrogen). Samples were analyzed using a nested filovirus family level consensus PCR (cPCR) targeting a 680 bp fragment of the filovirus L gene[11] (Round 1: Filo-MOD-FWD: TITTYTCHVTICAAAAICAYTGGG, FiloL.conR: AC CATCATRTTRCTIGGRAAKGCTTT; Round 2: Filo-MOD-FWD: TITTYTCHV TICAAAAICAYTGGG, Filo-MOD-RVS: GCYTCISMIAIIGTTTGIACATT), an Ebolavirus genus level cPCR targeting a 187 bp fragment of the NP gene[47] (Round 1: SudZaiNP1(+): GAGACAACGGAAGCTAATGC, SudZaiNP1(−): AACGGAA GATCACCATCATG; Round 2: SudZaiNP2(+): GGTCAGTTTCTATCCTTTGC, SudZaiNP2(−): CATGTGTCCAACTGATTGCC), a RT-PCR specific for Ebola virus (EBOV) virus targeting the L-gene[48] (EBOV FWD:AACTGATTTAGAGA AATACAATCTTGC, EBOV RVS: AATGCATCCAATTAAAAACATTC, Probe 1: FAM-ATTGCAACCGTTGCTATGGT-MGB, Probe 2: FAM-TAGAATATTGTA ACCGTTGCT-MGB) and a RT-PCR specific for the BOMV virus, targeting the L-gene[11] (Filo_UCD_qFor: TCTCGACGAAGGTCATTAGCGA, Filo_UCD_qRev: TTGCTCTGGTACTCGCTTGGT, Filo_UCD_probe: FAM-TGCTGGGATGCT GTCTTTGAGCCT-BHQ). Samples were analyzed for MARV using qRT-PCR targeting the VP35 gene. Bands of the expected size were excised from 1% agarose and purified using the Qiaquick kit (Qiagen Inc.). Purified PCR products were cloned (pCR4-TOPO vector; Invitrogen Corp.) and sequenced (ABI 3730 Capillary Electrophoresis Genetic Analyzer; Applied Biosystems, Inc., Foster City, CA). Libraries for genome sequencing were generated with the Kapa Hyper Library kit (Kapabiosystems, Roche)[32] and with VirCapSeq-VERT[33], and sequenced on the Illumina Miseq platform.

For each of the qRT-PCR positive bats from Kasewe, RNA was also purified using Tri-pure reagent (Invitrogen) and the Direct-Zol RNA Miniprep Plus kit (Zymo Research). NP and VP35 PCR products were obtained and sequenced on the SeqStudio Genetic Analyzer (Thermofisher Scientific).

RNA from the virus isolates was obtained and purified as described above and then used to perform whole genome sequencing on the MiSeq (Illumina). DNA sequencing libraries were constructed using the NEBNext rRNA Depletion kit (Human/Mouse/Rat) #E6310 and NEBNext Ultra II RNA Library Prep Kit for Illumina'' #E7335. Indexed libraries generated from pooled liver and spleen (Liv/ Spl) of bats 960 and 968, as well as axillary lymph node (LN) from bat 960 were pooled and run on the same flowcell using the MiSeq v2 reagent kit. Libraries from bat 1000 Liv/Spl were run on a separate flowcell.

**Phylogenetic analysis.** Sequences from the NP and VP35 genes were concatenated and a multiple sequence alignment was generated using the Clustal Omega program[49]. A maximum likelihood phylogeny was constructed using the best-fit nucleotide substitution model (GTR+G) in PhyML 3.0[50,51].

The reads from the virus isolates were mapped to the Angola 1379c reference strain (GenBank No.: DQ447653), and a consensus sequence was obtained for each using Geneious v 11.1.2. The consensus sequences were then aligned with all other MARV full-length sequences using the Clustal Omega program. A maximum likelihood phylogeny was constructed as described above using the best-fit nucleotide substitution model (GTR+I+G).

**Virus isolation and immunofluorescence assay.** All virus isolations were performed at the CDC under biosafety level 4 conditions. Tissue homogenates were placed into 500 µL DMEM/fungizone/penstrep (100 units/mL penicillin; 100 µg/ mL streptomycin; 2.50 µg/mL amphotericin B; Life Technologies) with 2% fetal calf serum[14]. The entire eluate was used to inoculate Vero-E6 cells (American Type Culture Collection, CRL-1586; mycoplasma-free) in 25 cm² flasks for 1 h at 37 °C and 5% CO₂. Maintenance media (DMEM containing 2% fetal bovine serum, 100 units/mL penicillin, and 100 µg/mL streptomycin) was then added to cultures; cells were monitored for 14 days with a media change on day 7.

All cultures were tested by immunofluorescence assay for MARV antigen at 7 and 14 DPI[14]. Immunofluorescence assay spot slides prepared from inoculated Vero E6 cells were fixed in acetone and then gamma-irradiated. After being incubated with a 1:100 dilution of rabbit anti-MARV polyclonal (in-house) or normal rabbit serum (negative control; in-house) for 30 min at 37 °C, rinsed two times with 1× PBS for 10 min, incubated with a 1:40 dilution of goat anti-rabbit fluorescein isothiocyanate (Capel-ICN Pharmaceuticals, Aurora, OH, USA) for 30 min at 37 °C, rinsed with 1× PBS for 7 min, stained with Eriochrome Black T (in-house) for 7 min and rinsed with 1× PBS for 7 min, the slides were observed under a fluorescence microscope.

**Serology.** Serum samples were tested at the CDC for the presence of MARV-specific IgG antibodies by indirect ELISA[52]. Wells of 96-well ELISA plates were coated (100 µL) with a 1:2000 dilution of MARV antigen lysate (in-house) and corresponding wells were coated with an equivalent dilution of uninfected control lysate (in-house). After incubation overnight at 4 °C, the plates were washed with PBS containing 0.1% Tween-20 (PBS-T) and 100 µL of serum diluent (PBS containing 5% skim milk and 0.1% tween-20) was added to each well of the plate. After 10 min, 33 µL of a 21:521 dilution of gamma-irradiated bat serum pre-diluted in masterplate diluent (PBS containing 5% skim milk powder, 0.5% tween-20 and 1%

thimerosal) was added to the first well of the plate and four-fold serial dilutions were performed. Final bat serum concentrations were 1:100, 1:400, 1:1600, and 1:6400. Following a 1 h incubation at 37 °C, the plates were washed with PBS-T and 100 μL of a 1:11,000 dilution of goat anti-bat IgG conjugated to horseradish peroxidase (Bethyl Laboratories, Montgomery, TX, USA) in serum diluent was added to the plates. The manufacturer notes that this antibody reacts specifically with bat IgG and with light chains common to other immunoglobulins. After incubation for 1 h at 37 °C, the plates were washed with PBS-T, 100 μL of the Two-Component ABTS Peroxidase System (KPL, Gaithersburg, MD, USA) was added, and the plates were allowed to incubate for 30 min at 37 °C. The plates were then read on a microplate spectrophotometer set at 410 nm. The optical density (OD) values of each four-fold serial dilution were visually inspected to ensure linearity. To negate non-specific background reactivity, adjusted OD values were calculated by subtracting the ODs at each four-fold dilution of wells coated with uninfected control antigen lysate from ODs at corresponding wells coated with MARV antigen lysate. The adjusted sum OD value was determined by summing the adjusted OD values at each four-fold serial dilution. A conservative threshold for MARV seropositivity of 0.92 was applied[52].

**Reporting summary**. Further information on research design is available in the Nature Research Reporting Summary linked to this article.

## Data availability

The authors declare that all data supporting the findings of this study are available within the article and its supplementary information files, or from the authors upon request. Genbank accession numbers for the Sierra Leone NP and VP35 sequences for all Kasbat SL 2017 and Kasbat SL 2018 sequences are as follows: MN193419—MN193431. Genbank accession numbers for the Sierra Leone full genome sequences for all Kasbat SL 2017 and Kasbat SL 2018 sequences are as follows: MN187403—MN187406. Genbank accession numbers for the SLAB3960Kakbat SL 2017 and SLAB410Koebat SL 2017 are MN258361—MN258362.

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

## Acknowledgements

We thank the government of Sierra Leone for permission to conduct this work; the Sierra Leone district and community stakeholders for support and for allowing us to perform sampling in their districts and communities; the residents and community leaders of both Tailu village (Kailahun District) and Kasewe (Moyamba District) for their support throughout the field work; the Ministry of Health and Sanitation and Ministry of Agriculture and Forestry district officers in the Moyamba, Kailahun, Koinadugu, and Kono districts; Ms. Mercy Mwaura; Brett Smith for laboratory and training assistance and Bridgette Smith for assistance with program management. Without the support and assistance of these people and institutions this work would not have been possible. This study was also made possible by the generous support of the American people through the United States Agency for International Development (USAID) Emerging Pandemic Threats PREDICT project (cooperative agreement number GHN-A-OO-09-00010-00), as well as support from the Centers for Disease Control and Prevention's Division of High Consequence Pathogens and Pathology and the Viral Special Pathogens Branch (cooperative agreement number NU50CK2019002475). The findings and conclusions in this report are those of the authors and do not necessarily represent the views of the Centers for Disease Control and Prevention or the Department of Health and Human Services.

## Author contributions

B.R.A., A.J.S., T.K.S., K.P., J.S.T. and A.L. wrote the paper. B.R.A., B.H.B., A.J.S., T.G., A.A.G., T.S., S.T.N., J.S.T. and A.L. provided critical review and editing of the paper. B.R.A., B.H.B., I.B., J.B., E.A., A.B., A.G., J.T., A.J.S., J.J., A.A.B., T.K.S., J.G., C.B., E.K., A.O., E.S., J.M., D.B., S.M.T.W., R.W., M.T., L.E., V.M-T, D.K., F.B., M.K., W.R., V.L., R.E.G.W., M.C., O.K., J.S.T. and A.L. conducted field work and data collection. A.H.K., A.J.S., J.J., T.K.S., I.C., I.F., J.G., A.T-B., M.B., J.D., A.C., V.O. and A.G. analyzed samples (PCR, serology, isolations). B.R.A., T.K.S., K.P., A.T-B., M.B., T.G. and B.H.B. analyzed data (sequencing, phylogenetics, statistics). C.T., T.G., J.A.K.M., J.S.T. and A.L. responsible for funding acquisition. A.A.G., A.J., S.M.K. provided in country support and permissions.

## Competing interests

The authors declare no competing interests.
