## [Peer Review File · Nature Communications]

Reviewers' Comments:

Reviewer #1:

Remarks to the Author:

This is a collaborative study between several partners including CDC, USAID PREDICT and local Universities in Sierra Leone. This collaboration led to the first detection of MARV in Rousette bats in West Africa. Previous field and experimental studies indicated that Rousettus are a reservoir of MARV and where targeted surveillance studies in this species was performed, MARV was detected but did not include West African bat populations. In this study the authors presented surveillance data that include antibody, virus isolation as well as genomic characterization of nucleic acids detected. Most studies will only report short sequences detected and the authors went further to characterize the viruses, isolate and also included serological results. Although the results are not surprising and correspond to what was detected in other regions, it certainly is a worthwhile contribution. The paper is well written, the methodology is well explained and the results are clear

I only have some minor comments below for improvement:

Introduction

Line 75-76; Gabon and Zambia re mentioned as countries where MARV was detected in ERBs despite no known associated human outbreaks. This is also true for South Africa. The MARV human cases identified was from Zimbabwe and did not originate in South Africa.

Results:

Line 100-102: The full genome sequences (Fig 3) were phylogenetically similar to the Angola-like MARV isolate. It does however form a separate group on the phylogenetic tree. The authors did not include a genome comparison and comparison of specific regions to indicate how similar it is. This can add value. The authors can also comment on how similar it is compared to other MARV strains e.g. Osolin compared to Uganda and DRC strains.

Line 151: Can the authors be more specific on the approximate size of the bat colonies observed?

Methods:

Line 210 -211: Was only Rousettus captured and tested or were other bat species also tested and it was negative? This will be important results to also include.

Line 212: Add GPS coordinates

Line 213: Add detail on PPE used when capturing and processing bats

Line 218: Add detail of the swabs used (Manufacturer and make)

Line 223: Add detail of the oral and rectal swabs used.

Line 225-227: The manuscript only describe sampling of Rousettus but the authors indicated that bat field IDs were confirmed with molecular methods. It is not clear why this was performed as Rousettus aegyptiacus is easy to morphologically identified and cannot be confused with other species occurring in the region.

Line 245 252: The methodology is not clear. It mentions real-time PCR in line 245 but then in Line 247-248 it describes first and second round PCR's. It then describes the amplicons that was generated, determining the concentration. In line 249 it then describe RNA extraction again followed by analysing amplicons (DNA) on the MinION platform. This is confusing

Line 260-262: The MinION platform are known to incorporate a high error rate. Was this considered when analysing the sequences and were sequences verified using other sequencing techniques with a lower error rate?

Table 1:

All bats that were positive were captured Sept, Oct, Dec. Were bats collected in other time periods that were negative and can the authors make any conclusions about seasonality of positives detected.

Sample type: It will add value to indicate if other sample types and swabs were also tested for the same bats and if it was negative

Sample 942 are indicated as positive but no sequence were obtained and organs are available. In

Table S1 sequences were obtained for 417 and 940 and these samples had a higher Ct value
The authors were able to obtain genome sequences from three isolates using the MiSeq and two non-isolates using Illumina. Was this easier achieved from the isolates compared to non-isolates?

Reviewer #2:

Remarks to the Author:

Dear editor,

I have reviewed the manuscript " Isolation of Angola-like Marburg virus from Egyptian rousette bats (*Rousettus aegyptiacus*) from West Africa. The manuscript describes the detection, isolation and phylogenetic analyses of Marburg virus (MARV) from Sierra Leone. The described data adds further evidence to the broad distribution of MARV throughout *Rousettus aegyptiacus* populations in Africa.

Given the relative relatedness to MARV detected in the largest outbreak to date it in Angolawould have been nice if the authors would have attempted some population genetic analyses on the host to look further into the relatedness between populations. Similar to what have been described for straw-colored fruit bats (<https://www.nature.com/articles/ncomms3770>).

Given that the authors strongly advocate the OneHealth approach, the novelty of this work is not just the detection of MARV in West Africa (which might have been expected), but the ability to design preemptive countermeasures. The statement "the clear and unwavering recommendation by the authors", is not really suggestive of a coordinated and strategic approach with the Sierra Leonian Ministry of Health. I would like to see a bit more in-dept discussion on how exactly this threat is currently assessed and countered (line 130). What levels of MoH and community engagements are set-up and how is this actually achieved, and what exact parameters are used to determine the levels of success of this approach.

Minor points:

Line 39: MVD, although this follows the apparently revisited nomenclature for disease syndromes caused by filoviruses, the inability to distinguish between EVD, MVD or any other febrile illness with associated multi organ failure makes the usages of this terminology rather useless in my opinion.

Line 79: One Health surveillance approach, it is a bit unclear what the authors are trying to say with this.

Consider revising table one, to include an overview of the results, or alternatively add one the the supplemental data to include sample numbers, age, sex, prevalence etc.

Responses to reviewer's comments. Manuscript NCOMMS-19-26164: Isolation of Angola-like Marburg virus in Egyptian rousette bats (*Rousettus aegyptiacus*) in West Africa.

Reviewer #1 (Remarks to the Author):

This is a collaborative study between several partners including CDC, USAID PREDICT and local Universities in Sierra Leone. This collaboration led to the first detection of MARV in Rousette bats in West Africa. Previous field and experimental studies indicated that Rousettus are a reservoir of MARV and where targeted surveillance studies in this species was performed, MARV was detected but did not include West African bat populations. In this study the authors presented surveillance data that include antibody, virus isolation as well as genomic characterization of nucleic acids detected. Most studies will only report short sequences detected and the authors went further to characterize the viruses, isolate and also included serological results. Although the results are not surprising and correspond to what was detected in other regions, it certainly is a worthwhile contribution. The paper is well written, the methodology is well explained and the results are clear

I only have some minor comments below for improvement:

Introduction

Line 75-76; Gabon and Zambia re mentioned as countries where MARV was detected in ERBs despite no known associated human outbreaks. This is also true for South Africa. The MARV human cases identified was from Zimbabwe and did not originate in South Africa.

We appreciate the comment and have added South Africa to the list along with an appropriate reference (Paweska et al., 2018, EID).

Results:

Line 100-102: The full genome sequences (Fig 3) were phylogenetically similar to the Angola-like MARV isolate. It does however form a separate group on the phylogenetic tree. The authors did not include a genome comparison and comparison of specific regions to indicate how similar it is. This can add value. The authors can also comment on how similar it is compared to other MARV strains e.g. Osolin compared to Uganda and DRC strains.

We agree with the reviewer's comment and have added supplemental tables 2-9 showing the nucleotide pairwise comparisons between all the Marburg virus sequences generated during this study with those of other well-known Marburg virus strains including Ozolins. The additional analyses include comparisons of whole genomes and each of the individual virus genes. The sentence "Genetic similarities between sequences are shown in supplemental Tables 2-9." was added to current line 119.

Line 151: Can the authors be more specific on the approximate size of the bat colonies observed?

We understand the reviewer's concern and would like to perform quantitative population estimates as already stated (line 194), "Future investigations will include mark-recapture studies to better estimate population sizes at these locations." Nevertheless, to clarify our qualitative population estimate, we have added the sentence (now line 192), "The determination that the ERB colonies are comparatively small is based on the lack of widespread fecal deposits on vegetation near the colony entrances, unlike the copious amounts normally seen in East African ERB populations."

Methods:

Line 210 -211: Was only *Rousettus* captured and tested or were other bat species also tested and it was negative? This will be important results to also include.

All bats captured at the locations described in the manuscript were tested for other filoviruses besides MARV. We have added the following to the text (now lines 93-97): “A total of 1755 bats from 42 species were captured and sampled from 4 districts in Sierra Leone: Moyamba (Kasewe Cave: 8°19'26.80"N, 12°10'36.00"W; n = 186), Kailahun (Tailu Village: 8°18'27.18"N, 10°30'55.32"W; n = 7), Koinadugu (Kakuya Cave: 9°41'24.00"N, 11°40'12.00"W; n = 131), and Kono (Koema Cave: 8°52'12.00"N, 10°48'36.00"W; n = 111) (Fig. 1). All bat samples were tested for 5 filoviruses (EBOV, TAFV, BDBV, MARV, and RAVV).”

Line 212: Add GPS coordinates

Latitude and longitude were added in the text for all 4 trapping sites – see previous response.

Line 213: Add detail on PPE used when capturing and processing bats

The proper use of PPE is described in the reference already noted (A chapter on methods for field collection of bat samples for Ebola testing). However, to add emphasis, we added the sentence (now lines 265-267), “When handling bats, all personnel wore appropriate personal protective equipment as previously described⁴⁵ that included disposable gowns, double gloves (including bite gloves if necessary), face shield and respiratory protection.”

Line 218: Add detail of the swabs used (Manufacturer and make)

Added Two oral secretion samples were taken using synthetic poly-tipped swabs (Fisher Scientific, Grand Island, NY, USA or Puritan, Guilford, ME, USA) to current lines 260.

Line 223: Add detail of the oral and rectal swabs used.

Added Two oral secretion samples were taken using synthetic poly-tipped swabs (Fisher Scientific, Grand Island, NY, USA or Puritan, Guilford, ME, USA) to current lines 260.

Line 225-227: The manuscript only describe sampling of *Rousettus* but the authors indicated that bat field IDs were confirmed with molecular methods. It is not clear why this was performed as *Rousettus aegyptiacus* is easy to morphologically identified and cannot be confused with other species occurring in the region.

The molecular IDs were done initially because some bats were field identified as possibly the smaller *Rousettus angolensis*. However, the molecular IDs indicated they were juvenile *Rousettus aegyptiacus*.

Line 245 252: The methodology is not clear. It mentions real-time PCR in line 245 but then in Line 247-248 it describes first and second round PCR's. It then describes the amplicons that was generated, determining the concentration. In line 249 it then describe RNA extraction again followed by analysing amplicons (DNA) on the MinION platform. This is confusing

We agree and have fixed the confusing description by clarifying real-time RT-PCR verses standard RT-PCR as well as removing one of the duplicate descriptions of the RNA extraction process.

Line 260-262: The MinION platform are known to incorporate a high error rate. Was this considered when analysing the sequences and were sequences verified using other sequencing techniques with a lower error rate?

VP35 and NP fragment sequences from only two of the 11 Marburg positive bats were determined by MinION. Because the standard RT-PCR amplifications were robust and created ample genetic material for sequencing, the coverage for these small fragments was very high, and therefore we have high confidence in the virus sequence. All the other sequences, including all full-length genome determinations, were performed using techniques with lower error rates.

Table 1:

All bats that were positive were captured Sept, Oct, Dec. Were bats collected in other time periods that were negative and can the authors make any conclusions about seasonality of positives detected.

At this point we have not sampled bats enough times at these locations to gain a clear understanding of seasonality and birth pulses at these ERB roost locations in Sierra Leone. Gaining an understanding of the ERB natural history in Sierra Leone is part of our ongoing effort. At Kasawe cave, attempts were made to capture bats in March 2018 during the height of the dry season, but without success. It remains unclear if the bats left the cave as the result of hunting pressure or a natural movement instigated by changes in local food supply (ERBs are fruit bats) or water. Of note, the bats were again present at Kasawe in Sept 2018, indicating there may be some seasonal movement.

Sample type: It will add value to indicate if other sample types and swabs were also tested for the same bats and if it was negative

The tissue taken from Kasewe Cave and Tailu Village bats during necropsy have been added to the materials and methods (now line 263). For bats collected at Kakuya and Koema Caves, the sample types are listed on current lines 269-270. Positive tissues are listed in Table 1. If a tissue is not listed there, it was negative for that bat.

Sample 942 are indicated as positive but no sequence were obtained and organs are available. In Table S1 sequences were obtained for 417 and 940 and these samples had a higher Ct value

The CT values for all of the samples mentioned are right on the edge of viral loads being good for producing a sequence. For whatever reason, the primers were unable to line up properly with the 942 sample and a sequence was not produced. The RNA in the sample may have been degraded due to freeze-thaw cycles or simply just fragmented to the point that sequencing did not work.

The authors were able to obtain genome sequences from three isolates using the MiSeq and two non-isolates using Illumina. Was this easier achieved from the isolates compared to non-isolates?

Yes, the higher viral loads that resulted from virus isolation were more amenable to sequencing on the MiSeq platform.

Reviewer #2 (Remarks to the Author):

Dear editor,

I have reviewed the manuscript “ Isolation of Angola-like Marburg virus from Egyptian rousette bats (*Rousettus aegyptiacus*) from West Africa. The manuscript describes the detection, isolation and phylogenetic analyses of Marburg virus (MARV) from Sierra Leone. The described data adds further evidence to the broad distribution of MARV throughout *Rousettus aegyptiacus* populations in Africa.

Given the relative relatedness to MARV detected in the largest outbreak to date it in Angola would have been nice if the authors would have attempted some population genetic analyses on the host to look further into the relatedness between populations. Similar to what have been described for straw-colored fruit bats (<https://www.nature.com/articles/ncomms3770>).

We agree with the reviewer that using population genetics to ascertain relatedness and mixing of ERB bat populations across the African continent would be useful information. The example provided by the reviewer showing panmixia in straw colored fruit bats is very interesting but represents a major effort by itself. Therefore, we feel that while likely insightful, a similar study of Egyptian rousette bats is beyond the scope of this investigation and would make an interesting and valuable contribution to the literature as a standalone study.

Given that the authors strongly advocate the OneHealth approach, the novelty of this work is not just the detection of MARV in West Africa (which might have been expected), but the ability to design preemptive countermeasures. The statement “the clear and unwavering recommendation by the authors”, is not really suggestive of a coordinated and strategic approach with the Sierra Leonian Ministry of Health. I would like to see a bit more in-depth discussion on how exactly this threat is currently assessed and countered (line 130). What levels of MoH and community engagements are set-up and how is this actually achieved, and what exact parameters are used to determine the levels of success of this approach.

We agree and the following was added to the text (now lines 151-170) to address Reviewer 2’s comment, “To accomplish this, a comprehensive One Health communications approach leveraging the human, animal, and environmental and emergency health sectors within the Ministries of Health and Sanitation and Agriculture, Forestry and Food Security along with other international partners was implemented across national, district, and local community levels. Through several engagement meetings with Ministry of Health and Sanitation and with several relevant ministries, departments and agencies, (Ministry of Agriculture Forestry and Food security, Ministry of Local Government, Ministry of Lands, Ministry of Mines and Mineral Resources, Environment and Protection Agencies, Office of National Security) over a two-week period, briefing documents including Marburg factsheets, MVD preparedness, detection and response plans were developed and presented at a national conference. This resulted in recommendations for outreach with the team comprising key stakeholders (government health and agriculture units, universities, development partners and district and local authorities) across the capital city and three of the districts (Moyamba, Koinadugu and Kono). This outreach team conducted initial information sharing events in each affected community followed by regular in-person meetings with traditional community leaders and other local stakeholders to provide key messages related to virus exposure risks and methods to reduce contact with bats. Concerns raised by local communities where bushmeat consumption brings them in contact with bats for livelihood were noted and discussed, and local perceptions about bats were explored in developing options for minimizing exposure

risks. As an additional national-level public preparedness measure, MVD has now been included in testing regimens at national laboratories in Sierra Leone.”

Minor points:

Line 39: MVD, although this follows the apparently revisited nomenclature for disease syndromes caused by filoviruses, the inability to distinguish between EVD, MVD or any other febrile illness with associated multi organ failure makes the usages of this terminology rather useless in my opinion.

We understand the reviewer’s opinion, but the two diseases, despite their similarity, are in fact caused by two different viruses from different filovirus genera.

Line 79: One Health surveillance approach, it is a bit unclear what the authors are trying to say with this.

The One Health approach refers to the interconnectedness between humans, animals and shared pathogens. Corresponding public health messaging includes, 1) health of humans (MOH community meetings and messaging), health of animals (leave the bats alone, promote conservation), health of environment (stay out of their habitat, promote bat conservation for health of forests – seed dispersal, pollination, etc.).

To clarify, starting on line 83, the text now reads, “Our findings highlight the value of engaging with all stakeholders with appropriate messaging that identify and mitigate pathogens of public health concern before recognized spillovers occur. This is in consonant with measures that ensure animal and environmental health. This underpins the One Health surveillance approach that recognizes the interconnected relationship between people and other organisms (plants and animals) in a shared environment.”

Consider revising table one, to include an overview of the results, or alternatively add one the supplemental data to include sample numbers, age, sex, prevalence etc.

A new supplemental table (Supplementary Table 10) was created showing the sample numbers, age, sex, and MARV active infection, isolation positive, and antibody results by sex and age cohort. The sentence “A summary of ERB qRT-PCR, virus isolation, and ELISA results by sex and age class is shown in supplemental Table 10.” was added to current lines 143-144.

Reviewers' Comments:

Reviewer #1:

Remarks to the Author:

The authors have addressed all the reviewer comments sufficiently. I have no more additional comments

Reviewer #2:

Remarks to the Author:

The authors have appropriately addressed the questions and concerns raised by the reviewers. the manuscript is suitable for publication